# Q-MAMBA: TOWARDS MORE EFFICIENT MAMBA MODELS VIA POST-TRAINING QUANTIZATION

## ABSTRACT

State Space Models (SSMs), such as Mamba, have recently demonstrated the potential to match or even surpass Transformers in language understanding tasks, making them a promising alternative for designing Large Language Models (LLMs). Concurrently, model quantization, particularly Post-Training Quantization (PTQ), has been proven effective in reducing memory usage and inference latency in LLMs. In this paper, we explore post-training quantization for Mamba (**Q-Mamba**) by converting both linear projections and state caches into low-bit integers for efficient inference. After a theoretical analysis of the causes of outliers in states, we propose **Decoupled Scale Quantization (DSQ)**, which mitigates outliers in both the state and channel dimensions by applying separate quantization scales. To preserve the selective ability of quantized Mamba, we introduce **Efficient Selectivity Reconstruction (ESR)**, a block-wise reconstruction method that involves a novel quantization simulation scheme, enabling fast parallel scan algorithms with the non-linear quantization function. We demonstrate the effectiveness of Q-Mamba across various quantization settings, model sizes, and both generation and zero-shot tasks. In particular, for Mamba2-2.7B with W8A8H4 quantization, Q-Mamba achieves a 50% reduction in memory consumption with only a 2.13% average accuracy degradation on zero-shot tasks.

## 1 INTRODUCTION

Large language models (LLMs), such as LLaMa (Touvron et al., 2023) and GPT-4 (OpenAI, 2023), have shown exceptional capabilities in general-purpose language understanding (Kaplan et al., 2020; Hoffmann et al., 2022). However, LLMs based on Transformer architectures still face a significant limitation: the computational cost of their attention mechanism scales quadratically with the sequence length (Vaswani et al., 2017). Therefore, prior works have focused on more efficient attention variants, such as structured state space models (SSMs) (Gu & Dao, 2023; Dao & Gu, 2024; Smith et al., 2023) and linear attention (Peng et al., 2023; Han et al., 2023; Child et al., 2019). Among these, the Mamba architecture (Gu & Dao, 2023; Dao & Gu, 2024) has been shown to match or exceed the downstream accuracy of Transformers on standard language modeling tasks (Waleffe et al., 2024). Following its success in natural language understanding, it has also garnered significant attention in other research areas, such as vision and multimodal tasks (Qiao et al., 2024; Zhu et al., 2024).

Like Transformers, Mamba language models also operate in two computation phases (Patel et al., 2024). The first is the prefill phase, where all input prompt tokens are processed in parallel through the model's forward pass to generate the first output token. During this phase, Mamba models (Gu & Dao, 2023; Dao & Gu, 2024) employ a hardware-efficient parallel algorithm to compute SSMs (Section 3). The second is the token generation phase, where subsequent output tokens are generated sequentially, relying on the cached state from previous tokens in the sequence. Due to the lack of computational parallelism, this phase tends to be more memory-bound and contributes significantly to the total generation latency.

Although Mamba has successfully replaced the $O(T^2)$ attention module with $O(T)$ selective state space models, our profiling results in Section 4 indicate that it still suffers from two inefficiencies during the generation stage. Firstly, similar to Transformers, the Mamba architecture consists of large linear layers, which require substantial GPU memory and slow down token generation

(Figure 2b). Secondly, as larger states allow more information to be stored, states in Mamba are expanded to be $N$ times larger than vanilla activations, where $N$ is the state dimension (128 in Mamba-2 models). Consequently, these state caches account for a significant portion of memory consumption, especially after quantizing weights to low bits (79.6% in Mamba2-2.7B with a batch size of 128, as shown in Figure 2a). In this paper, we address a key question: *Can Mamba models be further optimized through model compression techniques?*

In this paper, we propose **Q-Mamba**, which quantizes both **linear projections** and **state caches** into low-bit integers for Mamba models. Although previous research has successfully quantized Key and Value (KV) caches into low-bit representations in transformers (Liu et al., 2023; 2024b; Hooper et al., 2024), this work is the first to explore the quantization of state cache in Mamba architectures. We observe that states exhibit both outlier channels and outlier states (i.e., the state dimension contains large values across all channel dimensions), as shown in Figure 3. Further theoretical analysis reveals this phenomenon results from the computation of the outer products of two activations, each of which contains outliers in distinct dimensions. This observation motivates us to propose **Decoupled Scale Quantization (DSQ)**, which utilizes separate quantization scales for both dimensions. Additionally, the non-linear nature of the quantization function disrupts the original equivalence between recurrence and quadratic dual form, the latter being essential for efficient training. To address this, we propose Efficient Selectivity Reconstruction (ESR), which simulates quantization errors by quantizing only the final timestep during training. Specifically, ESR updates a small number of selective parameters (approximately 2% of the total) using just 128 training samples in a block-wise reconstruction manner.

Extensive experiments demonstrate that our methods achieve significant performance improvements for Mamba families on various evaluation metrics. To the best of our knowledge, we are the first to achieve W8A8H4 (8-bit linear projection and 4-bit states) for the Mamba architectures. For generation tasks, Q-Mamba achieved perplexities of 12.99 and 16.90 with 4-bit states on WikiText2 (Merity et al., 2017) and C4 (Pal et al., 2023), respectively, while baseline methods degraded to 21.18 and 29.86 even with 6-bit quantization. Additionally, Q-Mamba achieves W8A8H4 quantization for zero-shot tasks with only 2.13% and 2.11% average accuracy degradation on Mamba2-2.7B and Mamba2-1.3B, respectively.

## 2 RELATED WORKS

### 2.1 STATE SPACE MODEL

Transformer-based LLMs (Touvron et al., 2023; OpenAI, 2023) suffer from the computational cost of their attention mechanism scales quadratically with sequence length. Consequently, much research has focused on developing more efficient variants of attention, such as structured state space models (SSMs) (Gu & Dao, 2023; Dao & Gu, 2024; Smith et al., 2023). The original structured SSMs (S4) (Gu et al., 2022) were linear time-invariant (LTI) systems motivated by continuous-time online memorization. Many variants of structured SSMs have been proposed, for example, Gated SSM architectures, such as GSS (Mehta et al., 2023) and BiGS (Wang et al., 2023), incorporate a gating mechanism into SSMs for language modeling. Recently, the Mamba (Gu & Dao, 2023; Dao & Gu, 2024) architecture demonstrates promising performance on standard language modeling tasks (Waleffe et al., 2024), as well as on vision and multimodal tasks (Zhu et al., 2024; Qiao et al., 2024). Mamba showed that state expansion and selective ability are crucial for selecting and memorizing useful information in the hidden states.

### 2.2 MODEL QUANTIZATION

In the current era of burgeoning LLM development, model quantization has also become widely employed (Xiao et al., 2023; Lin et al., 2023; Frantar et al., 2022). Considering the substantial computational costs of retraining the entire model, much research has focused on Post-Training Quantization (PTQ), which requires only a small amount of calibration data to adjust a limited portion of the parameters. Typically, PTQ methods operate by quantizing and finetuning individual layers or small blocks of consecutive layers. For example, AdaRound (Nagel et al., 2020) uses gradient optimization to determine optimal rounding in a single convolution layer. For LLMs, previous quantization methods have identified significantly larger outliers in activations compared to

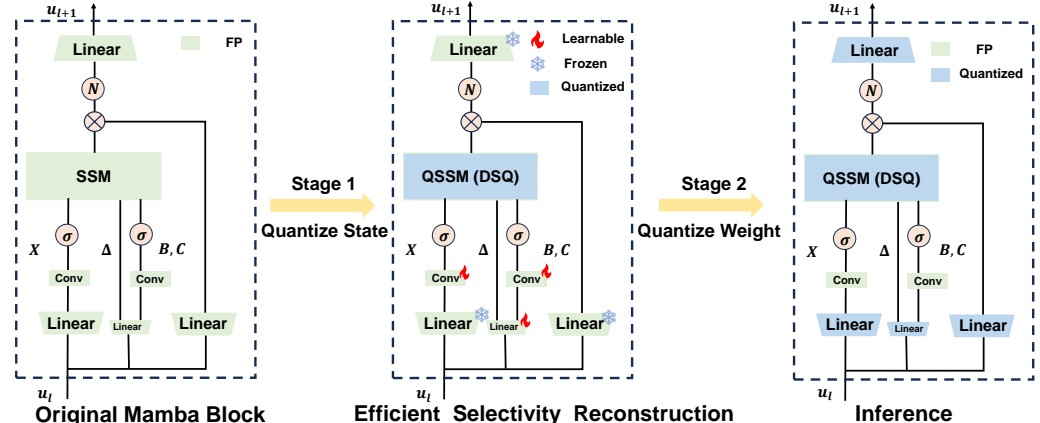

Figure 1: Schematic of the PTQ framework for Mamba. **Left**: The selective parameters $B$, $\Delta$, and $C$, along with the SSM inputs $x$, are generated by the input projections in the Mamba block. **Middle**: After quantizing states using DSQ, ESR updates a small number of selective parameters (approximately 2% of the total) in a block-wise reconstruction manner. **Right**: Finally, we quantize the linear projection into W8A8.

smaller convolutional neural networks (CNNs). To quantize both weights and activations into INT8, SmoothQuant (Xiao et al., 2023) mitigates activation outliers by shifting the quantization difficulty from activations to weights through a mathematically equivalent transformation. These outliers in activations also pose challenges even in scenarios where activations are not quantized (i.e., weight-only quantization) because they amplify the quantization errors of weights when multiplied with activations.

For Mamba models, states have an additional state dimension compared to standard activations, resulting in not only more significant memory consumption but also a distinctive distribution of outliers. To address this issue, we propose two novel methods that enable the quantization of states into 4-bit integers for the first time.

## 3 FOUNDATIONS

**State Space Model**. State space models (SSMs) in Equation (1) map a **1-dimensional** input sequence $x_t \in \mathbb{R}$ to an output sequence $y_t \in \mathbb{R}$ through a latent state $h_t \in \mathbb{R}^{(N,1)}$:

$$h_t = \bar{A}h_{t-1} + \bar{B}x_t \quad (1a) \qquad\qquad h'(t) = Ah(t) + Bx(t) \quad (2a)$$

$$y_t = Ch_t \quad (1b) \qquad\qquad y(t) = Ch(t) \quad (2b)$$

where $\bar{A} \in \mathbb{R}^{(N,N)}$, $B, \bar{B}, h_{t-1}, h_t, h(t) \in \mathbb{R}^{(N,1)}$, and $C \in \mathbb{R}^{(1,N)}$. Equation (1) can be viewed as discrete versions of a classical continuous system described by Equation (2). Specifically, a timescale parameter $\Delta$ is introduced to discretize the parameters $A$ and $B$ into their discrete counterparts, $\bar{A}$ and $\bar{B}$, as explained in the following sections.

**Mamba-1.** To operate on an input sequence $x_t$ **with $D$ channels**, rather than the scalar sequence described earlier, Mamba-1 (Gu & Dao, 2023) assumes that $\bar{A}$ has a diagonal structure and applies the SSM independently to each channel:

$$h_t = \bar{A} \odot h_{t-1} + \bar{B} \odot x_t, \qquad \bar{A}, \bar{B}, h_t, h_{t-1} \in \mathbb{R}^{(N,D)}, \qquad x_t \in \mathbb{R}^{(1,D)} \quad (3a)$$

$$y_t = Ch_t, \qquad\qquad C \in \mathbb{R}^{(1,N)}, \qquad y_t \in \mathbb{R}^{(1,D)} \quad (3b)$$

where $\odot$ denotes the element-wise product, with automatic broadcasting applied to dimensions of size one.. The discretized parameters are defined as $\bar{A} = \exp(A \odot \Delta)$ and $\bar{B} = B \odot \Delta$, where $A \in \mathbb{R}^{(N,D)}$, $B \in \mathbb{R}^{(N,1)}$, and $\Delta \in \mathbb{R}^{(1,D)}$. Unlike previous non-selective SSMs, Mamba set $\Delta$, $B$, and $C$ as functions of the inputs rather than fixed parameters. As a result, the variables $\bar{A}$, $\bar{B}$, and $C$ can vary across time steps to dynamically select relevant information from the context.

**Mamba-2.** To integrate the multi-head design of modern attention mechanisms into Mamba architectures, Mamba-2 (Dao & Gu, 2024) further assumes that $\bar{A}$ and $\bar{B}$ are identical across all dimensions **within the same head** where the head dimension $P \in \{64, 128\}$:

$$h_t = \bar{A} \cdot h_{t-1} + \bar{B} \otimes x_t, \qquad h_t, h_{t-1} \in \mathbb{R}^{(N,P)}, \qquad \bar{A} \in \mathbb{R}, \qquad \bar{B} \in \mathbb{R}^{(N,1)} \qquad (4a)$$

$$y_t = C h_t, \qquad\qquad\qquad C \in \mathbb{R}^{(1,N)}, \qquad x_t, y_t \in \mathbb{R}^{(1,P)} \qquad\qquad (4b)$$

The discretized parameters are still defined as $\bar{A} = \exp(A \odot \Delta)$ and $\bar{B} = B \odot \Delta$. However, unlike Mamba-1, $A$ and $\Delta$ are simplified into two scalars within a single head, transforming the operation between $\bar{B}$ and $x$ into an outer product. This simplification improves training efficiency and allows for a larger state size. Consequently, Mamba-2 increases the state size $N$ from 16 in Mamba-1 to 128. Figure 1 left shows the architecture of the Mamba-2 block. The selective parameters $B$, $\Delta$, and $C$, along with the SSM inputs $x_t$, are produced by the input projections in the Mamba block. Specifically, Mamba-2 employs $B = (u W_B)^\top, C = u W_C, \Delta = u W_\Delta, x_t = u W_x$, where $W_B, W_C \in \mathbb{R}^{(D,N)}, W_x \in \mathbb{R}^{(D,P)}, W_\Delta \in \mathbb{R}^{(D,1)}$ and $u \in \mathbb{R}^{(1,D)}$ represents the inputs of Mamba block.

**Parallel Training.** The recurrent mode described in Equation (1) is used only during the token generation phase, where output tokens are generated sequentially, relying on the cached state from the previous timestep. For parallel training, Mamba Dao & Gu (2024) establishes the **equivalence** between selective SSMs and semiseparable matrices, enabling the use of efficient algorithms for structured matrix multiplication (e.g, prefix sum algorithm (Goldberg & Zwick, 1995) ). Specifically, Equation (5) represents the **quadratic form** of Equation (1) to compute all timesteps simultaneously:

$$y_t = \sum_{s=0}^{t} C_t \bar{A}_{t:s}^\times B_s x_s, \quad \bar{B}, \bar{C}^\top \in \mathbb{R}^{(N,1)}, \quad \bar{A} \in \mathbb{R}^{(N,N)}, \quad x_t, y_t \in \mathbb{R}$$
$$y = Mx, \quad M_{ji} := C_j A_j \cdots A_{i+1} B_i, \qquad M \in \mathbb{R}^{(T,T)} \qquad\qquad (5)$$

where $M$ is N-semiseparable matrix.

This paper primarily focuses on quantizing the Mamba-2 architecture, which has demonstrated superior performance compared to Mamba-1 across various tasks (Waleffe et al., 2024; Dao & Gu, 2024). A detailed comparison between the two architectures from a quantization perspective is provided in the appendix. For more information on the Mamba architecture, please refer to the original papers (Gu & Dao, 2023; Dao & Gu, 2024).

## 4 ANALYSIS

In this section, we first analyze the memory consumption and runtime of primary components on the Mamba2-2.7B model, i.e., linear projection, 1D convolution, SSM, and LayerNorm. Based on the results presented in Figures 2a and Figures 2b, we can draw the following conclusions:

**Linear projections**. Similar to Transformers, large linear layers in Mamba not only require substantial GPU memory but also slow down token generation. When applying quantization to these linear layers, experiments in Section A.1 reveal that outliers exist in specific activation channels of Mamba, particularly in output projections. This phenomenon has also been observed in previous studies on Transformer-based LLMs (Xiao et al., 2023; Wei et al., 2022).

**States in SSMs**. As larger states allow more information to be stored, states in Mamba are expanded to be $N$ times larger than vanilla activations, where $N$ is the state dimension (128 in Mamba-2 models). Consequently, these state caches account for a significant portion of memory consumption, especially after quantizing weights to low bits (e.g., 79.6% in Mamba2-2.7B with a batch size of 128, as shown in Figure 2a). This phenomenon not only poses challenges for increasing the batch size to enhance throughput but also prevents further enlargement of state dimensions in Mamba models, which would improve their storage capacity for long contexts (Dao & Gu, 2024; Arora et al., 2024).

To address the above problems, in this paper, we aim to quantize both linear projections and state caches into low-bit integers for Mamba models.

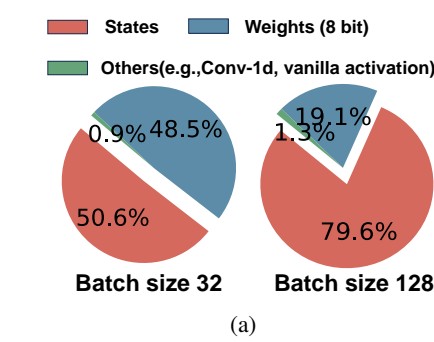 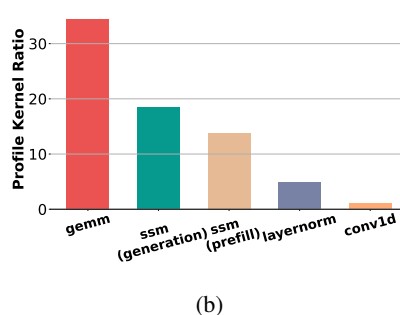

(a)                (b)

Figure 2: State distribution in Mamba2-370M. **Left**: Memory consumption of weights and state caches in Mamba2-2.7B with different batch sizes. **Right**: The Runtime of the Mamba2-2.7B model using NVIDIA profiling tools, with both prompt and generation lengths set to 100 and a batch size of 32.

## 5 METHOD

### 5.1 DECOUPLED SCALE QUANTIZATION

#### 5.1.1 OUTLIERS IN STATES

For Transformers, particularly LLMs, extensive research (Wei et al., 2022; Xiao et al., 2023; Liu et al., 2024a) has shown that the presence of outliers extends the range of activation values, which in turn increases quantization errors for normal values. In Mamba models, we observe a similar or even more pronounced issue with outliers in the states. As illustrated in the state distribution visualization in Figure 3(a), outliers are present in both state dimensions (red row) and channel dimensions (green column). Consequently, either per-channel quantization (i.e., using a different quantization step for each channel) or per-state quantization (i.e., using a different quantization step for each state) tends to ignore outliers in the other dimension. As shown in Table 3, the model's performance declines significantly when adopting the above quantization granularity, which calls for a more effective quantization method to address the problem.

#### 5.1.2 DECOUPLED SCALE QUANTIZATION

Motivated by the distribution characteristics shown in Figure 3, we present the following theorem, which reveals the underlying causes of this distribution and provides insights for a solution.

**Theorem 1.** *Assuming $u_t \sim \mathcal{N}(\mathbf{0}, \sigma \mathbf{I}_n)$ and $A_t$ is a constant, $B_t = (uW_B)^\top$, $x_t = uW_x$, the variance of states $h_t = A_t \cdot h_{t-1} + B_t \otimes x_t$ can be factorized into two vectors:*

$$Var[h_t] \propto \alpha \cdot \beta^T, \quad \alpha_i = ||W_{i,:}^x||_2^2 \quad and \quad \beta_i = ||W_{i,:}^B||_2^2 \tag{6}$$

The above theorem demonstrates that outliers in the channel dimension $P$ and state dimension $N$ can be attributed to variables $x_t$ and $B_t$, respectively. A visualization of this phenomenon is provided in Figure 3(b). This motivates us to propose a novel quantization scheme called **Decoupled Scale Quantization (DSQ)**, which utilizes separate quantization scales for the state dimension and the channel dimension:

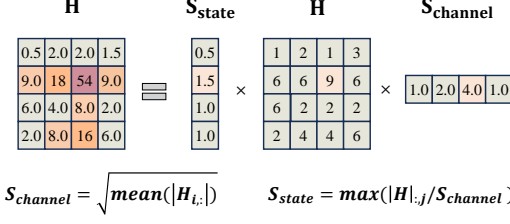

Figure 4: An illustration of DSQ.

$$Q(h) = \lfloor \frac{h}{S_{channel} \cdot S_{state}^\top} \rceil \odot (S_{channel} \cdot S_{state}^\top) \tag{7}$$

where $S_{channel} \in \mathbb{R}^P$, $S_{state} \in \mathbb{R}^N$ and $\lceil \cdot \rfloor$ denotes rounding floating-point values to the nearest integers, while $\odot$ signifies element-wise multiplication.

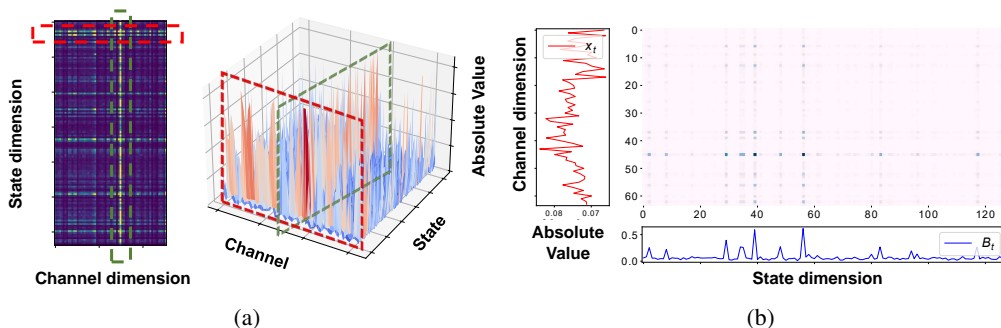

(a)                                               (b)

Figure 3: State distribution in Mamba2-370M. **Left**: Outliers exist in both specific state dimensions (red) and channel dimensions (green). **Right**: Further analysis reveals outliers in channel dimension and state dimension can be attributed to variables $x_t$ and $B_t$, respectively.

In this paragraph, we discuss how to compute scales given a specific state. To increase the effective quantization bits, both state and channel scales should accurately represent the magnitude of their respective dimensions. Therefore, an intuitive metric to determine these scales is the vector norm, such as maximum norm ($\| \cdot \|_\infty$) and $L^1$ norm ($\| \cdot \|_1$). However, in practice, we find that both norms result in even worse performance (see Table 5). Further visualization in Figure 8 shows that these norms are highly sensitive to outliers, resulting in even greater bit wastage. Therefore, for the channel scale, we use the square root of the mean values, which offers a more robust metric that mitigates the influence of outliers. After mitigating most outliers by smoothing the states with channel scale, we employ the MinMax method to compute state scale, which effectively compresses the data range and reduces information loss during quantization:

$$S_{channel,i} = \text{sqrt}(\text{mean}(\text{abs}(h_{i,:}))) = \sqrt{||h_{i,:}||_1} \tag{8}$$

$$S_{state,j} = \max(\text{abs}(\frac{h_{:,j}}{S_{channel}})) = ||\frac{h_{:,j}}{S_{channel}}||_\infty \tag{9}$$

where $i$ and $j$ denote subscripts indexing into the channel and state dimensions, respectively. Table 3 demonstrates that DSQ achieves negligible overhead while significantly improving performance.

## 5.2 EFFICIENT SELECTIVITY RECONSTRUCTION

To mitigate the performance loss caused by quantization, PTQ methods often apply block-wise reconstruction (Nagel et al., 2020; Li et al., 2021) with a few data. However, these methods cannot be directly applied to Mamba models due to the following differences: First, when applying the non-linear quantization function to states $h_t$, the definition of SSMs can no longer be reformulated into quadratic mode for parallel training. Second, given the distinct mechanisms between Mamba and Transformers, it is necessary to investigate which set of parameters is critical for restoring model performance and which may lead to overfitting. In this section, we will present **Efficient Selectivity Reconstruction** (ESR) with the mechanisms to address these two challenges in Section 5.2.1 and Section 5.2.2, respectively.

### 5.2.1 QUANTIZATION-AWARE STATE SPACE MODEL

To minimize memory bandwidth utilization, we store state caches as low-bit elements, then load and dequantize them before computation at the next timestep. This process defines a new sequence transformation through the quantized latent state $h_t^q$ in Equation (10). It is important to distinguish $h_t^q = \bar{A}Q(h_{t-1}^q) + \bar{B}x_t$ from the quantized value of the original $h_t$, denoted as $Q(h_t)$, where the latter is given by $Q(h_t) = Q(\bar{A}h_{t-1} + \bar{B}x_t)$.

$$h_t^q = \bar{A}Q(h_{t-1}^q) + \bar{B}x_t, \tag{10a}$$

$$y_t^q = Ch_t^q \tag{10b}$$

A significant challenge arises because the original parallel training algorithms are incompatible with the quantization scenario. Specifically, the non-linear nature of the quantization function breaks the equivalence between the recurrent and quadratic modes. (In other words, this equivalence relies on the linearity of original SSMs.) A naive approach would involve directly applying Equation (10) for token-by-token generation. However, given the large input lengths (e.g., 2048), this method is extremely slow and impractical. Therefore, to apply block-wise reconstruction for Mamba models, it is essential to first investigate how to effectively simulate quantization errors during training.

$$
\begin{aligned}
h_t^q &= \bar{A}_t Q(h_{t-1}^q) + \bar{B}_t x_t \\
&= \bar{A}_t Q(\bar{A}_{t-1} h_{t-2}^q + \bar{B}_{t-1} x_{t-1}) + \bar{B}_t x_t \\
&\neq \bar{A}_t \bar{A}_{t-1} Q(h_{t-2}^q) + \bar{A}_t \bar{B}_{t-1} x_{t-1} + \bar{B}_t x_t \\
&\neq \sum_{s=1}^{t} \bar{A}_s \bar{A}_{s+1} \cdots \bar{A}_t \bar{B}_s x_s
\end{aligned}
\tag{11}
$$

To gain insight into this problem, we focus on the difference between the quantized and original states, which is defined as $\delta_t = h_t^q - h_t$. By substituting $\delta_t$ into Equation (10), we observe that $\delta_t$ is composed of two parts: the quantization error propagated from the previous timestep, $\delta_{t-1}$, and the quantization error introduced in the current timestep:

$$
\begin{aligned}
\delta_t = h_t^q - h_t &= \bar{A}_t Q(h_{t-1}^q) + \bar{B}_t x_t - (\bar{A}_t h_{t-1} + \bar{B}_t x_t) \\
&= \bar{A}_t \cdot (Q(h_{t-1}^q) - h_{t-1}) \\
&= \bar{A}_t \cdot (Q(h_{t-1} + \delta_{t-1}) - h_{t-1})
\end{aligned}
\tag{12}
$$

Assuming that quantization errors $\delta_{t-1}$ are sufficiently small compared to the hidden state $h_{t-1}$, we discard $\delta_{t-1}$ and focus only on the quantization errors at the current timestep:

$$
\begin{aligned}
Q(h_{t-1} + \delta_{t-1}) &\approx Q(h_{t-1}) + Q'(h_{t-1}) \cdot \delta_{t-1} \approx Q(h_{t-1}) \\
&\implies h_t^q \approx \bar{A}_t Q(h_{t-1}) + \bar{B}_t x_t
\end{aligned}
\tag{13}
$$

Equation (13) enables us to utilize the parallel algorithm to compute $h_t$ at all timesteps, then simulate the quantization errors by quantizing only one step during training. In the appendix, we present the pseudocode for the parallel training of quantization-aware SSMs for illustrative purposes. Table 4 demonstrates the effectiveness of this quantization simulation, especially in low-bit settings.

### 5.2.2 SELECTIVITY GUIDED ADAPTATION

In the Mamba block, the selective parameters $B$, $\Delta$, and $C$, along with the SSM inputs $x_t$, are generated through input projections, as shown in Figure 1. During block-wise reconstruction, we freeze the linear projections corresponding to the SSM inputs $x$ and $z$, while keeping the linear projections for selective parameters $B$, $C$, and $\Delta$ learnable, which is referred to as Selectivity Guided Adaptation (SGA) (Figure 1, middle). Specifically,

$$\min_{\{W_v^q | v \in B, C, \Delta\}} \left\| \mathcal{B}_l(W_v^{FP}, h_t^{FP}; u_l) - \mathcal{B}_l(W_v^q, h_t^q; u_l) \right\|_2, \quad v \in \{x, z, B, C, \Delta\} \tag{14}$$

where $B_l$ denotes the $-th$ mapping function for the $l$-th Mamba block and $u_l$ represents the block's inputs. $W^{FP}$ and $W^q$ represent the weights of the original model and the quantized model, respectively.

SGA offers two primary advantages: First, the success of Mamba is largely attributed to the selectivity of parameters $\bar{A}$, $\bar{B}$, and $\bar{C}$, which distinguishes it from earlier non-selective SSMs (Gu et al., 2020; Smith et al., 2023). Thus, we hypothesize that this selectivity also plays a critical role in maintaining performance after quantization. Second, SGA reduces the number of learnable parameters, mitigating the risk of overfitting with limited calibration data. For example, in Mamba2-2.7B, learnable parameters account for only about 2% of the total. Note that during this fine-tuning process, the linear layers remain in floating-point values and can be quantized afterward (Figure 1, right).

Table 1: Evaluation results of the Mamba-2 models on generation tasks. #W, #A, and #H indicate weight bits, activation bits, and state bits, respectively.

| Bits | Method | WikiText2 ↓ | | | | | C4 ↓ | | | | |
|------|--------|------|------|------|------|------|------|------|------|------|------|
| | | 130M | 370M | 780M | 1.3B | 2.7B | 130M | 350M | 780M | 1.3B | 2.7B |
| FP16 | - | 20.04 | 14.16 | 11.81 | 10.42 | 9.06 | 22.25 | 16.95 | 14.66 | 13.27 | 11.95 |
| W16A16H4 | Baseline | 976.56 | 913.34 | 865.78 | 1556.15 | 116.23 | 542.048 | 599.49 | 911.31 | 529.55 | 96.93 |
| | **Q-Mamba** | **45.73** | **22.24** | **19.07** | **15.20** | **11.55** | **39.46** | **26.36** | **22.45** | **19.14** | **14.90** |
| W16A16H6 | Baseline | 249.09 | 134.91 | 38.04 | 23.62 | 13.60 | 322.97 | 101.75 | 38.24 | 23.73 | 19.61 |
| | **Q-Mamba** | **23.79** | **15.33** | **12.69** | **11.37** | **9.59** | **25.11** | **18.27** | **15.66** | **14.52** | **12.57** |
| W16A16H8 | Baseline | 20.97 | 14.83 | 12.04 | 10.52 | 9.11 | 22.97 | 17.45 | 14.85 | 13.40 | **12.01** |
| | **Q-Mamba** | **20.49** | **14.26** | **11.86** | **10.51** | **9.11** | **22.64** | **17.05** | **14.73** | **13.39** | 12.04 |
| W8A8H4 | Baseline | 2024.49 | 1013.15 | 7225.39 | 6375.57 | 364.84 | 635.86 | 795.28 | 10716.17 | 2788.23 | 298.57 |
| | **Q-Mamba** | **53.12** | **27.53** | **23.53** | **17.60** | **12.99** | **46.90** | **32.91** | **26.79** | **21.56** | **16.90** |
| W8A8H6 | Baseline | 357.69 | 220.09 | 96.51 | 47.28 | 21.18 | 526.59 | 171.90 | 79.70 | 40.46 | 29.86 |
| | **Q-Mamba** | **26.75** | **17.27** | **14.51** | **13.05** | **10.84** | **28.18** | **20.53** | **17.79** | **16.45** | **14.46** |
| W8A8H8 | Baseline | 23.60 | 16.69 | 14.32 | **11.85** | 10.42 | 25.51 | 19.50 | 17.44 | **14.86** | 13.73 |
| | **Q-Mamba** | **22.88** | **15.83** | **13.57** | 11.93 | **10.36** | **25.01** | **18.84** | **16.80** | 15.03 | **13.69** |

# 6 EXPERIMENTS

## 6.1 EXPERIMENT SETUP

**Settings.** We conduct experiments on the Mamba-2 (Dao & Gu, 2024) models across various model sizes (130M, 370M, 780M, 1.3B, 2.7B). We initialize quantized models using a full-precision model. We utilize the AdamW optimizer with zero weight decay to optimize the learnable parameters in ESR. The learning rate for learnable parameters is set to 1e-3. RedPajama is an open-source reproduction of the pre-training data for LLaMA(Touvron et al., 2023). We employ a calibration dataset consisting of 128 randomly selected 2048-token segments from the RedPajama (Computer, 2023) dataset, except for Mamba2-2.7B, which utilizes 256 samples. The entire training process is facilitated on a single NVIDIA A800 GPU, using a batch size of 1 over 3 epochs. For linear projections, we apply SmoothQuant (Xiao et al., 2023) with per-token quantization. For state quantization, we use INT8, INT6, and INT4 schemes (e.g., W8A8H4 refers to 8-bit linear projection and 4-bit quantization of the states). We utilize MinMax per-channel quantization (introduced in Section 5.1.2) as state quantization **baseline**.

**Evaluation Tasks.** We evaluate our methods on both language generation and zero-shot tasks. We report the perplexity on WikiText2 (Merity et al., 2017) and C4 (Pal et al., 2023). For zero-shot tasks, we provide accuracy on datasets including PIQA (Bisk et al., 2020), ARC (Clark et al., 2018), BoolQ (Clark et al., 2019), OpenBookQA (Mihaylov et al., 2018), HellaSwag (Zellers et al., 2019) and Winogrande (Sakaguchi et al., 2020).

## 6.2 MAIN RESULTS

**Generation Tasks.** We evaluate generation tasks in recurrent mode with a sequence length of 2048. The results in Table 1 demonstrate the effectiveness of Q-Mamba across various quantization configurations. For INT8 state quantization, we exclusively utilize DSQ without ESR, as DSQ alone achieves nearly lossless quantization compared to full-precision models. Without our methods, states are limited to 8-bit quantization, with lower-bit quantization, such as 6-bit, leading to significant performance degradation, e.g., 23.62 perplexity for Mamba2-1.3B on the WikiText2 dataset. In contrast, Q-Mamba facilitates nearly lossless 6-bit quantization, achieving a minimal degradation of only 0.53 perplexity for Mamba2-2.7B and 0.88 perplexity for Mamba2-1.3B. Moreover, Q-Mamba enables effective 4-bit quantization and is compatible with the linear projection quantization approach. For example, Q-Mamba achieves 12.99 perplexity in W8A8H4 quantization settings for the Mamba2-2.7B model.

**Zero-shot Tasks.** We evaluate the performance of Q-Mamba on zero-shot tasks using the lm-eval-harness (Gao et al., 2024) framework in Table 2. Q-Mamba significantly improves the average accuracy across various models. For example, it increases the average accuracy by 6.37%, 6.55%,

Table 2: Evaluation results of the Mamba-2 models with W8A8H4 (8-bit weights, activations, and 4-bit states) on zero-shot tasks.

| Model | Method | OBQA | PIQA | ARC-E | ARC-C | HellaSwag | WINO | AVG ↑ |
|-------|--------|------|------|-------|-------|-----------|------|-------|
| Mamba2-130M | FP | 30.6 | 64.9 | 47.4 | 24.2 | 35.3 | 52.1 | 42.41 |
| | Baseline | **30.8** | **63.4** | 45.6 | **24.6** | **34.1** | 51.93 | **41.73** |
| | **Q-Mamba** | 30.0 | 63.0 | **45.7** | 23.4 | 33.9 | **53.3** | 41.55 |
| Mamba2-370M | FP | 32.4 | 70.5 | 54.9 | 26.9 | 46.9 | 55.7 | 47.83 |
| | Baseline | 28.6 | 58.6 | 46.5 | 24.9 | 30.4 | 53.0 | 40.34 |
| | **Q-Mamba** | **32.8** | **68.4** | **53.8** | **26.7** | **43.8** | **54.8** | **46.71** |
| Mamba2-780M | FP | 36.2 | 72.0 | 61.0 | 28.5 | 54.9 | 60.2 | 52.13 |
| | Baseline | 32.0 | 61.8 | 50.5 | 25.9 | 29.5 | **57.5** | 42.85 |
| | **Q-Mamba** | **34.2** | **69.6** | **57.3** | **27.6** | **52.1** | 55.6 | **49.4** |
| Mamba2-1.3B | FP | 37.8 | 73.2 | 64.3 | 33.3 | 59.9 | 60.9 | 54.9 |
| | Baseline | **35.6** | 67.1 | 57.6 | 29.2 | 36.8 | 58.5 | 47.46 |
| | **Q-Mamba** | 34.8 | **72.6** | **62.5** | **31.4** | **55.7** | **59.5** | **52.77** |
| Mamba2-2.7B | FP | 38.8 | 76.4 | 69.6 | 36.4 | 66.6 | 64.0 | 58.63 |
| | Baseline | 39.8 | 73.2 | 66.8 | **36.0** | 56.4 | 59.6 | 55.30 |
| | **Q-Mamba** | **40.0** | **73.9** | **66.8** | 35.4 | **62.0** | **61.0** | **56.52** |

and 5.31% on the 370M, 780M, and 1.3B models. Additionally, for Mamba2-2.7B and Mamba2-1.3B, Q-Mamba achieves W8A8H4 quantization with only 2.13% and 2.11% accuracy degradation.

Table 3: The performance and overheads of different quantization methods on Mamba2-370M. $P$ and $N$ denote channel and state dimensions, respectively.

| Granularity | WikiText2 ↓ | Overheads |
|-------------|-------------|-----------|
| Per-tensor | 4815.83 | $\frac{1}{P \times N}$ |
| Per-channel | 3364.58 | $\frac{1}{P}$ |
| Per-state | 947.88 | $\frac{1}{N}$ |
| **DSQ** | **25.73** | $\frac{1}{P} + \frac{1}{N}$ |

Table 4: Efficacy of each component in ESR. ESR enables adjusting parameters of Mamba blocks after quantizing states **in block-wise reconstruction**. When combined with SGA, these two techniques further enhance performance.

| Method | WikiText2 ↓ | C4 ↓ |
|--------|-------------|------|
| DSQ w/o ESR | 25.73 | 29.94 |
| DSQ+ESR (w/o SGA) | 23.73 | 28.19 |
| **DSQ+ESR (w/ SGA)** | **21.92** | **25.99** |

## 6.3 ABLATIONS

In this section, we conduct experiments to validate the efficacy of each component, as well as the design choices for DSQ, training epochs, and calibration data size. In Section A.3 of the Appendix, we provide visualizations of DSQ and a detailed analysis of the impact of trainable parameters in ESR.

**Effectiveness of each component.** Table 3 demonstrates that DSQ is essential in state quantization. The model's performance declines significantly when per-channel or per-state quantization methods are adopted. By decoupling scales in the state and channel dimensions, DSQ mitigates outliers in both dimensions with negligible overhead. Table 4 shows that we can further enhance model performance in block-wise reconstruction with ESR. Furthermore, finetuning selective parameters instead of all parameters can help avoid overfitting and yield better results.

**Design choices of DSQ.** The results in Table 5 highlight the critical importance of selecting appropriate quantization scales for DSQ. Firstly, squaring the norms as quantization scales is essential for maintaining stability. Furthermore, using mean values yields superior performance compared to relying on maximum values.

**Samples and epochs for block-wise reconstruction.** To ensure training efficiency, we set 3 epochs and 128 samples for all experiments, except for Mamba2-2.7B, where we use 256 samples. How-

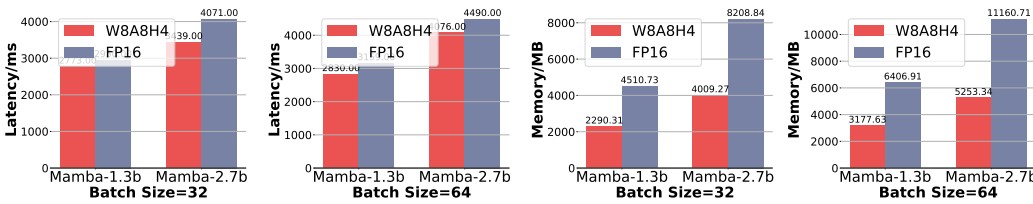

Figure 5: Inference latency and memory usage of the Mamba2 models with different batch sizes on NVIDIA GeForce RTX 3090.

ever, as shown in Figure 6, performance can be further improved by increasing the number of training samples and epochs.

## 6.4 EFFICIENCY

Figure 5 presents the memory and time requirements for inference using Mamba2 models. For W8A8 linear projections, we employ CUDA INT8 GEMM, following the approach of SmoothQuant (Xiao et al., 2023). For INT4 state quantization, we implement SSM kernels with quantized and packed states with Triton (Tillet et al., 2019), a language and compiler for CUDA computation. Both the input context and generation length are set to 100. The results show that the quantized models can reduce memory usage by half while maintaining or even improving inference latency.

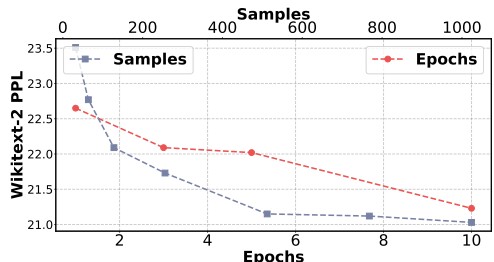

Figure 6: Illustration of WikiText2 perplexity of W16A16H4 quantization with different training samples and epochs.

Table 5: Impact of different design choices for DSQ. Experiments are conducted on Mamba2-370M with W16A16H4 quantization.

| Method | WikiText2 ↓ | C4 ↓ |
|---|---|---|
| abs.max | inf | inf |
| abs.max.sqrt | 42.88 | 46.61 |
| abs.mean | inf | inf |
| **abs.mean.sqrt** | **25.73** | **29.94** |

## 7 CONCLUSION

In this paper, we propose Q-Mamba, a novel quantization scheme designed for Mamba models. After visualizing outliers in states, we conduct a theoretical analysis of their causes and propose Decoupled Scale Quantization (DSQ). By decoupling scales in the state and channel dimensions, DSQ mitigates outliers in both dimensions while introducing negligible overhead. To further boost performance through block-wise reconstruction, we propose Efficient Selectivity Reconstruction (ESR), which includes a novel quantization simulation method that enables efficient fine-tuning of selective parameters with parallel scan mode. We validate the performance of Q-Mamba across various quantization settings, model sizes, and both generation and zero-shot tasks. In conclusion, Q-Mamba demonstrates that Mamba architectures have the potential for further optimization when combined with other model compression techniques.

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

# A APPENDIX

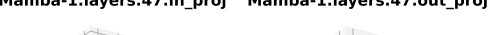

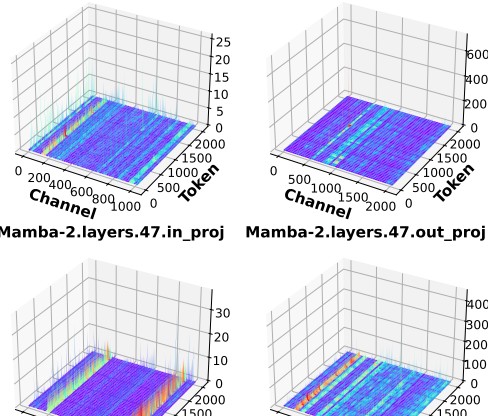

Figure 7: Visualization of inputs for linear projections. The out projection suffers from more severe outliers compared to the in projection.

## A.1 PREVIOUS PTQ METHODS ON MAMBA

In Section 4, we analyze the quantization of linear projections in Mamba models. Here, we provide more detailed results about previous PTQ methods on Mamba-1 and Mamba-2 models. We will analyze the difference between Mamba-1 models and Mamba-2 models from a view of model quantization. The results presented in Table 6 indicate that Mamba2 models exhibit greater robustness to quantization compared to Mamba1 models. Further analysis in Figure 7 reveals that this improvement is largely due to the additional LayerNorm applied before the output projection in Mamba2, which helps to reduce outliers to a certain extent. Moreover, this LayerNorm simplifies the implementation of previous PTQ methods based on smoothing between weights and activations, such as SmoothQuant (Xiao et al., 2023) and AWQ (Lin et al., 2023). As a result, this paper primarily focuses on Mamba2 models, which not only feature larger state dimensions but are also more amenable to quantization.

| Model | Method | WikiText2 | C4 |
|---|---|---|---|
| Mamba1-370M | FP | 14.31 | 17.23 |
| | W8A8 | 18.95 | 23.04 |
| | W8A8+SQ | 16.17 | 19.85 |
| | W4A16+ GPTQ | 16.03 | 19.06 |
| Mamba2-370M | FP | 14.16 | 16.95 |
| | W8A8 | 17.14 | 20.10 |
| | W8A8+SQ | 15.71 | 18.72 |
| | W4A16+GPTQ | 15.81 | 18.71 |

Table 6: Different PTQ methods for Mamba models. Mamba-1 models suffer much more serious outliers in output projections because of the absence of LayerNorm before it.

## A.2 PROOF

**Theorem 2.** *Assuming $u_t \sim \mathcal{N}(\mathbf{0}, \sigma \boldsymbol{I}_n)$ and $A_t$ is a constant, $B_t, x_t = split(Wu_t)$ ($B_t \in \mathbb{R}^N$, $x_t \in \mathbb{R}^P$), the variance of states $h_t$ can be factorized into two vectors:*

$$h_t = A_t \cdot h_{t-1} + x_t \cdot B_t^\top \tag{15}$$

$$Var[h_t] \propto \alpha \cdot \beta^T, \quad \alpha_i = ||W_{i,:}^x||_2^2 \quad and \quad \beta_i = ||W_{i,:}^B||_2^2 \tag{16}$$

where $\alpha \in \mathbb{R}^P$ and $\beta \in \mathbb{R}^N$ and $W^B, W^x = split(W, dim = 0)$

*Proof.* Firstly, we can reformulate Equation (**??**) as a prefix sum:

$$h_t = \sum_i^t A_{i:t} x_i B_i^\top, \quad where \quad A_{i:t} = A_i \times A_{i+1} \times \ldots A_t \tag{17}$$

Then, we can compute the mean of states $h_t$ as follows:

$$
\begin{aligned}
\mathbb{E}[h_t] &= \sum_i^t A_{i:t} \mathbb{E}[x_i B_i^\top] \\
&= \sum_i^t A_{i:t} \mathbb{E}[W^x u_i u_i^\top W^{b^\top}] \\
&= \sum_i^t A_{i:t} W^x \mathbb{E}[u_i u_i^\top] W^{b^\top} \\
&= \sum_i^t A_{i:t} \sigma W^x W^{b^\top}
\end{aligned} \tag{18}
$$

After computing the mean of the states, we can similarly compute the variance of the states $h_t$. The equality (a) is attributed to Lemma 1.

$$
\begin{aligned}
\mathrm{Var}[x_i B_i^\top] &= \mathbb{E}[(W^x u_i u_i^\top W^{b^\top} - \sigma W^x W^{b^\top})] \\
&= \mathbb{E}[(W^x (u_i u_i^\top) W^{b^\top})^2] - 2\sigma \cdot \mathbb{E}[W^x W^{b^\top} \odot (W^x u_i u_i^\top W^{b^\top})] + (\sigma W^x W^{b^\top})^2 \\
&\overset{(a)}{=} \sigma^2 \alpha \cdot \beta^\top + 2\sigma^2 \cdot (W^x W_b^\top)^2 - 2\sigma^2 \cdot (W^x W_b^\top)^2 + \sigma^2 \cdot (W^x W^{b^\top})^2 \\
&= \sigma^2 \alpha \cdot \beta^\top + \sigma^2 \cdot (W^x W^{b^\top})^2
\end{aligned} \tag{19}
$$

Here, we assume that the second term $(W^x W^{b^\top})^2$ is sufficiently small compared to $\alpha \cdot \beta^\top$, and then we obtain:

$$\mathrm{Var}[h_t] = \quad = (\sigma^2 \sum_i^t A_{i:t}) \cdot (\alpha \cdot \beta^\top) \tag{20}$$

$\square$

**Lemma 1.** *Assuming $z \sim \mathcal{N}(\mathbf{0}, \mathbf{I}_n)$, $w_1, w_2 \in \mathbb{R}^n$, we have the following conclusions:*

$$\mathbb{E}[(w_1^\top z)^2 (w_2^\top z)^2] = ||w_1||_2^2 \cdot ||w_2||_2^2 + 2(w_1^\top w_2)^2 \tag{21}$$

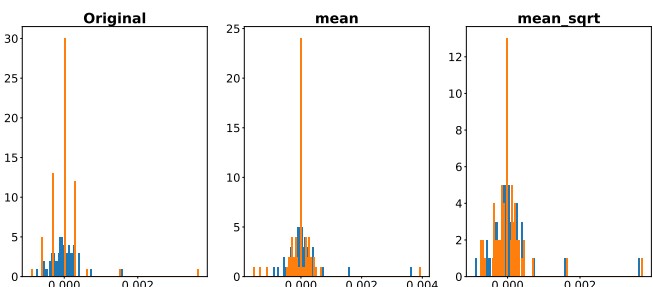

Figure 8: An illustration of how DSQ enhances performance.

*Proof.* Let $A$ and $B$ be two arbitrary symmetric matrices, we have:

$$
\begin{aligned}
\mathbb{E}\left[x^\top A x \cdot x^\top B x\right] &= \mathbb{E}\left[\sum_{i,j} x_i a_{ij} x_j \sum_{k,l} x_k b_{kl} x_l\right] \\
&= \mathbb{E}\left[\sum_{i,k} a_{ii} b_{kk} x_i^2 x_k^2 + 4\sum_{i<j} a_{ij} b_{ij} x_i^2 x_j^2\right] \\
&= \sum_{i,k} a_{ii} b_{kk} + 2\sum_i a_{ii} b_{ii} + 2\left(\sum_{i,j} a_{ij} b_{ij} - \sum_i a_{ii} b_{ii}\right) \\
&= \sum_i a_{ii} \sum_k b_{kk} + 2\sum_{i,j} a_{ij} b_{ij} \\
&= \text{Tr}(A)\text{Tr}(B) + 2\text{Tr}(AB)
\end{aligned}
\tag{22}
$$

A special case occurs when $A = w_1 w_1^\top$ and $B = w_2 w_2^\top$:

$$
\mathbb{E}[(w_1^\top z)^2 (w_2^\top z)^2] = ||w_1||_2^2 \cdot ||w_2||_2^2 + 2(w_1^\top w_2)^2
\tag{23}
$$

$\square$

Although this theorem imposes strict constraints on the SSM inputs $u_t$ (Gaussian distribution) and $A_t$ (constant), it sufficiently reveals the following fact: outliers in the channel dimension $P$ and state dimension $N$ can be attributed to the variables $x_t \in \mathbb{R}^{(T,P)}$ and $B_t \in \mathbb{R}^{(T,N)}$, respectively. Figure 3 provides a visualization of this phenomenon.

### A.3 MORE ABLATION STUDIES

**Visualization of DSQ.** Figure 8 illustrates how DSQ improves performance. The presence of outliers causes MinMax quantization to waste a significant portion of available quantization slots, resulting in large rounding errors. Although introducing channel scales $S_{channel}$ helps make the quantization slots non-uniform, the mean norm remains sensitive to outliers, even unexpectedly amplifying them (as shown in the middle figure).

**Trainable parameters in ESR.** Table 7 demonstrates the effectiveness of our choice of trainable parameters in ESR: Fine-tuning selective parameters ($B$, $C$, and $\Delta$), layer normalization, and convolution yields the best perplexity. In contrast, including $x$ and $z$ results in worse performance. We attribute this to the fact that fine-tuning all parameters can lead to overfitting and necessitates end-to-end training.

### A.4 PSEUDOCODE

In this section, we present the pseudocode for the parallel training of quantization-aware SSMs. To enhance understanding, we also include the pseudocode for the recurrent and quadratic modes of

| Norm | $\Delta$,B,C,D | Conv-1D | X,Z | WikiText2 | C4 |
|:---:|:---:|:---:|:---:|:---:|:---:|
| | | | | 25.73 | 29.94 |
| ✓ | | | | 24.76 | 29.02 |
| | ✓ | | | 23.27 | 27.22 |
| | | ✓ | | 25.24 | 29.09 |
| | | | ✓ | 24.99 | 28.88 |
| ✓ | ✓ | | | 22.51 | 27.00 |
| ✓ | | ✓ | | 24.93 | 28.87 |
| ✓ | | | ✓ | 25.31 | 29.43 |
| | ✓ | ✓ | | 22.68 | 26.91 |
| | ✓ | | ✓ | 22.97 | 26.41 |
| | | ✓ | ✓ | 25.66 | 28.89 |
| ✓ | ✓ | ✓ | | **21.92** | **25.99** |
| ✓ | ✓ | | ✓ | 23.63 | 27,43 |
| ✓ | | ✓ | ✓ | 24.89 | 29.04 |
| | ✓ | ✓ | ✓ | 23.01 | 26.98 |
| ✓ | ✓ | ✓ | ✓ | 23.73 | 28.19 |

Table 7: The performance of W16A16H4 quantization for Mamba2-370M with different trainable parameters in the ESR.

Mamba-2. It is worth noting that these pseudocodes are provided solely for illustrative purposes and do not represent actual implementations.

```python
def ParallelSSM(
    A,  # bsz * num_head * len
    B,  # bsz * num_head * len * state_dim
    C,  # bsz * num_head * len * state_dim
    x   # bsz * num_head * len * channel_dim
):
    BC = C @ B.transpose(-1, -2)
    prefix_sum = torch.cumsum(A)

    # L : bsz * num_head * len * len
    L = torch.tril(prefix_sum.unsqueeze(-1) - prefix_sum.unsqueeze(-2))

    ABC = L * BC
    y = ABC @ x
    return y
```

```python
def RecurrentSSM_onestep(
    A,  # bsz * num_head
    B,  # bsz * num_head * state_dim
    C,  # bsz * num_head * state_dim
    x,  # bsz * num_head * channel_dim
    last_state  # bsz * num_head * channel_dim * state_dim
):
    current_state = A * last_state + B.unsqueeze(-2) * x.unsqueeze(-1)
    output = current_state @ C.unsqueeze(-1)
    return output.squeeze(-1)
```

```python
def QuantizationAwareParallelSSM(
    A,  # bsz * num_head * len
    B,  # bsz * num_head * len * state_dim
    C,  # bsz * num_head * len * state_dim
    x   # bsz * num_head * len * channel_dim
):
    BX = B.unsqueeze(-2) * x.unsqueeze(-1)
    prefix_sum = torch.cumsum(A)
```

```
9      L = torch.tril(prefix_sum.unsqueeze(-1) - prefix_sum.unsqueeze(-2))
10     state = torch.einsum('bhldn,bhll->bhldn', BX, L)
11
12     # Simulate the quantization errors at the last timestep
13     # Error case: qstate =  fake_quant(state)
14     qstate = A[:, :, 1:] * fake_quant(state)[:, :, :-1] + BX[:, :, 1:]
15
16     y = torch.einsum('bhldn,bhln->bhld', qstate, C)
17     return y
```

