# OpenReview forum: "Q-Mamba: Towards more efficient Mamba models via Post-Training Quantization"
_ICLR.cc/2025/Conference — Submitted to ICLR 2025_

### Official Review · Reviewer_Aixt · 2024-10-26

**Soundness:** 2
**Presentation:** 2
**Contribution:** 1
**Rating:** 3
**Confidence:** 4

**Summary:**

The authors aim to investigate the unique challenges of quantizing state space models. Their investigation highlights the importance of outlier reduction, which motivates their proposal for fine-grained scaling factor selection (i.e., decoupled scale quantization) and block-wise reconstruction (i.e., efficient selectivity reconstruction).

**Strengths:**

The majority of quantization research on language models focus on the standard transformer architecture, namely Llama models, leaving state space models an under-explored model architecture within the space of post-training quantization. Given the importance of state space models, the topic is relevant and timely.

**Weaknesses:**

Outliers are a well-established challenge in neural network quantization, not just in language models but across neural architectures. It is unclear the unique challenges that SSMs pose over standard transformer architectures. While the authors claim this is the first paper to study SSM quantization, it is unclear where the unique challenges arise. Enough prior studies have investigated fine-grained scaling factors and block-wise reconstruction to warrant benchmarking against those techniques. Ultimately, it is unclear if existing techniques are already sufficient to solve the challenges the authors present. For example, many PTQ works have investigated groupwise quantization (e.g., GPTQ) to increase the granularity. This also establishes a grid of scaling factors similar to the decoupled scale quantization proposal. Finally, rotation-based quantization techniques have been studied in a handful of papers over the last year (e.g., QuIP, QuaRot, SpinQuant). There is no mention of any of these approaches in the paper, and it has been one of the most effective techniques to reducing outliers in language models.

**Questions:**

The notation in Equation 1 confuses me. The matrix dimensions don't seem to line up. Should there be a transpose for $A_t h_{t-1}$ since $A_t$ is $N \times N$ and $h_{t -1}$ is $T \times N$? Also, I imagine $B_t x_t$ is an outer product and would also need to be transposed since $x_t$ is $T \times P$ and $B_t$ is $N \times 1$? If this understanding is correct, can you please explain where the $N \times P$ comes in?

It is unclear why BRECQ (or other existing block-wise reconstruction algorithms) can't be applied to SSMs. The analysis claims parallel training and layer selection are the challenges, but it is not clear if (or how) this is a restriction on the algorithm or some implementation of it. Can you please provide more detail?

---

> ### Author Response · Authors · 2024-11-26
>
> **Q1:** Outliers are a well-established challenge in neural network quantization, not just in language models but across neural architectures. It is unclear the unique challenges that SSMs pose over standard transformer architectures. While the authors claim this is the first paper to study SSM quantization, it is unclear where the unique challenges arise. Enough prior studies have investigated fine-grained scaling factors and block-wise reconstruction to warrant benchmarking against those techniques. Ultimately, it is unclear if existing techniques are already sufficient to solve the challenges the authors present. For example, many PTQ works have investigated groupwise quantization (e.g., GPTQ) to increase the granularity. This also establishes a grid of scaling factors similar to the decoupled scale quantization proposal. Finally, rotation-based quantization techniques have been studied in a handful of papers over the last year (e.g., QuIP, QuaRot, SpinQuant). There is no mention of any of these approaches in the paper, and it has been one of the most effective techniques to reducing outliers in language models.
>
> **R1**:In this study, we emphasize that Q-Mamba focuses on the quantization of **state caches** in Mamba architectures, setting it apart from previous research on quantizing transformers and CNNs. States in Mamba are expanded to be $N$-times larger than standard activations, where $N$ represents the state dimension (128 in Mamba-2 models). Section 4 demonstrates that these state caches contribute significantly to memory consumption, particularly after weights are quantized to low-bit representations.  It is important to note that the quantization of linear projections for Mamba models is not the primary contribution of this work, although we present preliminary experiments on Mamba-1 and Mamba-2 using GPTQ and SmoothQuant. On the other hand, Q-Mamba is orthogonal and can be combined with previous quantization methods for linear layers, such as OmniQuant, AWQ, and Mamba-PTQ.
>
> **Q2:** It is unclear why BRECQ (or other existing block-wise reconstruction algorithms) can't be applied to SSMs. The analysis claims parallel training and layer selection are the challenges, but it is not clear if (or how) this is a restriction on the algorithm or some implementation of it. Can you please provide more detail?
>
> **R2: We emphasize that adjusting parameters for quantized states is a non-trivial task in block-wise reconstruction.**  SSMs (and other linear attention mechanisms) rely on the equivalence between two computational modes: the quadratic mode for training and the recurrent mode for inference. This equivalence depends on the linear nature of the hidden-to-hidden transformations, $h_t = A_t h_{t-1} + B_t x_t$. In contrast, RNNs, defined by $h_t = \sigma(A_t h_{t-1} + B_t x_t)$ (where $\sigma$ is a nonlinear activation function), cannot be parallelized during training. To minimize memory bandwidth utilization, we store state caches as low-bit elements and subsequently load and dequantize them before computation at the next timestep. This process introduces a new sequence transformation, defined as $h_t^q = \bar{A} Q(h_{t-1}^q) + \bar{B} x_t$, where $Q(\cdot)$ is a non-linear quantization function. It is crucial to distinguish $h_t^q = \bar{A} Q(h_{t-1}^q) + \bar{B} x_t$ from the quantized value of the original $h_t$, i.e., $Q(h_t) = Q(\bar{A} h_{t-1} + \bar{B} x_t)$. A naive approach would directly apply recurrent mode for token-by-token generation in the training. However, this method becomes computationally prohibitive for large input lengths (e.g., 2048). To address this, we propose Efficient Selectivity Reconstruction (ESR), which utilizes the parallel algorithm to compute $h_t$ across all timesteps and simulates quantization errors by quantizing only a single step during training.
>
> **Q3:** The notation in Equation 1 confuses me. The matrix dimensions don't seem to line up. Should there be a transpose for \( A_t h_{t-1} \) since \( A_t \) is \( N \times N \) and \( h_{t-1} \) is \( T \times N \)? Also, I imagine \( B_t x_t \) is an outer product and would also need to be transposed since \( x_t \) is \( T \times P \) and \( B_t \) is \( N \times 1 \). If this understanding is correct, can you please explain where the \( N \times P \) comes in?
>
> **R3:**We apologize for the confusion. We have significantly modified the background section.

---

> > ### Comment · Reviewer_Aixt · 2024-11-26
> >
> > It is still unclear what unique challenges arise when quantizing state caches. You point to the volume of state caches, their outliers, and recurrence. For volume, the standard transformer architecture can also have a large volume of weights and/or activations depending on model architecture and inference stage (prefill vs. decode). For outliers, activation quantization is typically more challenging as models increase in size (see this ICML paper [1]). Finally, recurrence is not a new challenge for post-training quantization. There is plenty of existing work on RNNs, and now diffusion models [2, 3, 4].
> >
> > How can you justify your baselines when techniques have been proposed to tackle these challenges? Enough prior studies have investigated fine-grained scaling factors and block-wise reconstruction to warrant benchmarking against those techniques, even if they didn't benchmark on Mamba architectures. Ultimately, it is unclear if existing techniques are already sufficient to solve the challenges that you present.
> >
> > Also, please note that line 818 has an undefined equation reference.
> >
> > [1] Li, Shiyao, et al. "Evaluating quantized large language models." arXiv preprint arXiv:2402.18158 (2024).
> >
> > [2] Shang, Yuzhang, et al. "Post-training quantization on diffusion models." Proceedings of the IEEE/CVF conference on computer vision and pattern recognition. 2023.
> >
> > [3] Li, Xiuyu, et al. "Q-diffusion: Quantizing diffusion models." Proceedings of the IEEE/CVF International Conference on Computer Vision. 2023.
> >
> > [4] So, Junhyuk, et al. "Temporal dynamic quantization for diffusion models." Advances in Neural Information Processing Systems 36 (2024).

---

> > > ### Author Response · Authors · 2024-11-28
> > >
> > > We understand that the quantization of Transformer models is critically important and has garnered significant attention in recent years. However, considering the outstanding performance of Mamba models on natural language tasks and their O(T) computational complexity, we believe that exploration on accelerating Mamba models is also valuable to the community.
> > >
> > > For **volume**, states in Mamba models are expanded to be N-times larger than standard **activations**, where N represents the state dimension (128 in Mamba-2 models). Section 4 demonstrates that these state caches significantly contribute to memory consumption, especially after **weights are quantized to low-bit representations**. Furthermore, **we emphasize that quantizing states and quantizing linear layers are not mutually exclusive. Our approach incorporates both, achieving quantization for Mamba models with W8A8H4.**
> > >
> > > For **outliers**, our contribution extends beyond merely identifying their presence in states, which complicates quantization. **We provide a detailed analysis of their distribution characteristics and investigate the underlying causes of their emergence.**
> > >
> > > For **recurrence**, the primary distinction between RNNs and Mamba lies in their differing training algorithms. Structured state-space models (SSMs) employ a quadratic mode during training and a recurrent mode during inference, whereas RNNs depend on backpropagation through time (BPTT), which cannot be parallelized. Consequently, RNN models are prohibitively slow when processing sequences as long as 2048 tokens and cannot scale to deeper architectures. **As noted in the official response to reviewers, the lack of equivalence between the two modes in Mamba presents a challenge for block-wise reconstruction.** Regarding diffusion models, while they also involve timesteps, they primarily employ CNNs (as evident from the papers you cited).
> > >
> > > For **DSQ**, the baseline methods use per-channel quantization, and the effective group size is already set to 128 (the state dimension). Regarding block-wise reconstruction, we have explained why previous methods cannot be directly applied to the sequential quantization of states. If you have further questions, please let us know.

---

> > > > ### Comment · Reviewer_Aixt · 2024-12-02
> > > >
> > > > Thank you for your responses. I will keep my score.

---

> > > > > ### Author Response · Authors · 2024-12-03
> > > > >
> > > > > We respect your opinion and appreciate your feedback. We would like to understand which aspects have not been addressed, as this would be very helpful for further improvements to the paper.

---

### Official Review · Reviewer_tXAm · 2024-10-28

**Soundness:** 2
**Presentation:** 2
**Contribution:** 2
**Rating:** 3
**Confidence:** 3

**Summary:**

This paper introduces Q-Mamba, a post-training quantization (PTQ) approach aimed at improving the efficiency of Mamba models by converting linear projections and state caches into low-bit integers. The authors present two techniques: Decoupled Scale Quantization (DSQ), which addresses the outliers in both state and channel dimensions, and Efficient Selectivity Reconstruction (ESR) to preserve the model’s selectivity. They demonstrate Q-Mamba’s efficacy across various quantization settings and tasks, reporting a 50% reduction in memory with minimal accuracy degradation in some configurations.

**Strengths:**

1. **Relevance**: The paper addresses a key issue by optimizing Mamba’s memory and computational requirements, making it relevant to the field of efficient large language models (LLMs).

2. **Novel Quantization Scheme**: DSQ’s approach to outlier mitigation in both state and channel dimensions adds a unique element to quantization in SSMs.

3. **Practical Impac**: Q-MAMBA's claimed memory reduction is substantial, especially for real-world applications where memory efficiency is essential.

4. **Clarity in Methodology**: The authors provide a theoretical foundation for DSQ and clear motivation for ESR, contributing to understanding their method.

**Weaknesses:**

1- **Computation Overhead in Quantization**: The quantization and dequantization process requires computing an outer product of two vectors, which can be computationally expensive compared to other group quantization methods available in the field.

2- **Lack of Comprehensive Evaluation**: The experimental results lack sufficient diversity across tasks and model sizes. Evaluating Q-Mamba’s performance on additional Mamba family models would strengthen its claims of versatility.

3- **Limited Baseline Comparisons**: The paper could benefit from comparisons with related quantization methods like OmniQuant [1], OPTQ [2], AWQ [3], and Mamba-PTQ [4]. Assessing Q-Mamba against these methods would provide better insights into its performance, especially given that Mamba is composed largely of linear layers for which these quantization schemes have proven effective.

4- **Insufficient Analysis of Latency**: The paper does not sufficiently explore the impact of Q-Mamba on inference latency, which could be critical for real-world deployment. More timing results can enrich the experiments section further.

5- **Potential Misstep in DSQ Calculation**: In Equation 7, it appears that the mean of the absolute values of elements in $h$ should be used rather than the mean. If this is not an oversight, additional explanation would be valuable.

6- **Justification of ESR Design Choices**: ESR’s design choices, particularly which parameters are made learnable, lack sufficient exploration, and further analysis on this aspect would clarify ESR's effectiveness.


[1] Shao et al. "OmniQuant: Omnidirectionally Calibrated Quantization for Large Language Models", ICLR 2024

[2] Frantar et al. "OPTQ: Accurate Quantization for Generative Pre-trained Transformers", ICLR 2023

[3] Lin et al. "AWQ: Activation-aware Weight Quantization for LLM Compression and Acceleration", MLSys 2024

[4] Pierro et al. "Mamba-PTQ: Outlier Channels in Recurrent Large Language Models", ICML 2024

**Questions:**

1- **Comparison to Related Work**: How does Q-MAMBA compare to other quantization approaches such as Mamba-PTQ, OmniQuant, OPTQ, and AWQ in terms of performance and memory efficiency? The papers are cited in the weaknesses section.

2- **Impact on Latency**: What is Q-Mamba's effect on real-world latency, especially for larger batch sizes or latency-sensitive tasks? Additionally, what are the overheads of the outer products for the quantization and dequantizaiton of tensors?

3- **Scalability of DSQ**: Can DSQ be effectively scaled or applied to other architectures, or does it require specific adjustments for compatibility?

4- Clarification on DSQ Calculation: In Equation 7, should the mean of the absolute values of elements in $h$ be used instead of the mean? Further clarification on this point would be helpful.

---

> ### Author Response · Authors · 2024-11-26
>
> **Q1:**  **Computation Overhead in Quantization**. The quantization and dequantization process requires computing an outer product of two vectors, which can be computationally expensive compared to other group quantization methods available in the field.
>
> **R1**: During the generation stage, state caches $h_t$ are quantized at each time step to reduce memory consumption and are subsequently dequantized in the next step, which is similar to KV cache quantization. We implement a cuda kernel that load quantized $h_t{-1}$, compute SSM $h_t=A_th_{t-1}+B_tx_t$, then quantize and pack $h_t$ and store into memory.  The computation overheads for scales are neligiable compared to original SSMs computation and reduce half memory-bandwith utlizaition.
>
> **Q2:Lack of Comprehensive Evaluation**: The experimental results lack sufficient diversity across tasks and model sizes. Evaluating Q-Mamba’s performance on additional Mamba family models would strengthen its claims of versatility.
>
> The results of different tasks (including generation tasks and zerot-shot tasks) and model sizes (from 130M to 2.7B) are shown in Table 1 and Table 2.
>
> **Q3:Limited Baseline Comparisons**: The paper could benefit from comparisons with related quantization methods like OmniQuant [1], OPTQ [2], AWQ [3], and Mamba-PTQ [4]. Assessing Q-Mamba against these methods would provide better insights into its performance, especially given that Mamba is composed largely of linear layers for which these quantization schemes have proven effectiv
>
> **R3:**In this study, we emphasize that Q-Mamba focuses on the quantization of **state caches** in Mamba architectures, setting it apart from previous research on quantizing transformers and CNNs. States in Mamba are expanded to be $N$-times larger than standard activations, where $N$ represents the state dimension (128 in Mamba-2 models). Section 4 demonstrates that these state caches contribute significantly to memory consumption, particularly after weights are quantized to low-bit representations.  **It is important to note that the quantization of linear projections for Mamba models is not the primary contribution of this work,** although we present preliminary experiments on Mamba-1 and Mamba-2 using OPTQ[2] and SmoothQuant[5]. On the other hand, Q-Mamba is orthogonal and can be combined with previous quantization methods for linear layers, such as OmniQuant[1], AWQ,[3] and Mamba-PTQ[4].
>
> [1] Shao et al. "OmniQuant: Omnidirectionally Calibrated Quantization for Large Language Models", ICLR 2024
>
> [2] Frantar et al. "OPTQ: Accurate Quantization for Generative Pre-trained Transformers", ICLR 2023
>
> [3] Lin et al. "AWQ: Activation-aware Weight Quantization for LLM Compression and Acceleration", MLSys 2024
>
> [4] Pierro et al. "Mamba-PTQ: Outlier Channels in Recurrent Large Language Models", ICML 2024
>
> [5] Xiao et al. "SmoothQuant: Accurate and Efficient Post-Training Quantization for Large Language Model", ICML, 2023

---

> > ### Author Response · Authors · 2024-11-26
> >
> > **Q4:Insufficient Analysis of Latency:** The paper does not sufficiently explore the impact of Q-Mamba on inference latency, which could be critical for real-world deployment. More timing results can enrich the experiments section further.
> >
> > **R4:** Figure 6 presents the memory and time requirements for inference using Mamba2 models with batch size 32 and 64.
> >
> > **Q5:Potential Misstep in DSQ Calculation**: In Equation 7, it appears that the mean of the absolute values of elements in h should be used rather than the mean. If this is not an oversight, additional explanation would be valuable.
> >
> > **R5:** To increase the effective quantization bits, quantization scales should accurately represent the magnitude of the quantized values. Thus, the mean of the absolute values of elements is a common metric for computation scales to avoid cancellation between positive and negative values. For example, the mean of the absolute values in a normal Gaussian distribution is zero, which clearly is not an optimal scale.
> >
> > **Q6:Justification of ESR Design Choices**: ESR’s design choices, particularly which parameters are made learnable, lack sufficient exploration, and further analysis on this aspect would clarify ESR's effectiveness.
> >
> > **R6**:SGA offers two primary advantages: First, the success of Mamba is largely attributed to the selectivity of parameters $\bar{A}$, $\bar{B}$, and $\bar{C}$, which distinguishes it from earlier non-selective SSMs.
> > Thus, we hypothesize that this selectivity also plays a critical role in maintaining performance after quantization. Second, SGA reduces the number of learnable parameters, mitigating the risk of overfitting with limited calibration data.For example, in Mamba2-2.7B, learnable parameters account for only about 2\% of the total. The ablations studies in Table 7 and Figure 7 demonstrates its effectiveness.
> >
> > **Q7:Scalability of DSQ**: Can DSQ be effectively scaled or applied to other architectures, or does it require specific adjustments for compatibility?
> >
> > **R7:** DSQ can be applied to any tensor with more than one dimension, but its performance is optimal when the outliers in the tensor can be separated along different dimensions, as stated in Theorem 1.

---

### Official Review · Reviewer_UHAm · 2024-11-01

**Soundness:** 3
**Presentation:** 2
**Contribution:** 1
**Rating:** 6
**Confidence:** 5

**Summary:**

The paper introduces the first scheme for post-training quantization (PTQ) of Mamba models. In contrast to transformers, which are well studied in the context of PTQ, Mamba models have large state tensors that pose unique quantization challenges, as they can take a significant portion of memory, especially after quantizing weights.
They tackle this issue by focusing on the quantization of the states cache to low bits, down to 4 bits. They begin by examining the activation tensors and find that they have large outliers across both channel and state dimensions, making quantization difficult. They propose two methods to address this:

1. Decoupled Scale Quantization (DSQ) that uses different scales and quantization schemes along the channel and state dimensions. This flattens outliers across both dimensions

2. Efficient Selective Reconstruction (ESG), which is a local fine-tuning technique for certain SSM parameters ($B$, $C$ and $\Delta$).

By applying SmoothQuant to the rest of the linear layers of the network and ESG+DSQ to states the quantize the Mamba-2 models to W8A8H4. Specifically for Mamba2-2.7B they achieve 50% memory reduction with only 2.17% drop in accuracy degradation for zero-shot tasks

**Strengths:**

* The authors are the first to tackle the quantization of Mamba models and achieve impressive compression of the state tensors that can take a large proportion of memory. They also show the efficacy of their work by showing the memory reduction and latency improvement in Figure 6.

* The authors systematically study the state tensors and the presence of outliers to motivate Decoupled Scale Quantization with good visualization in Fig. 4 & Fig 8.

* The profiling of Mamba inference and memory requirements supported by figures (Fig, 2 & Fig. 8)

* An easy-to-read paper with good ablation studies

**Weaknesses:**

Unfortunately, the paper lacks innovation from a quantization perspective even if the application of PTQ is quite novel, recycling ideas that are already well established in existing quantization literature. I believe that this paper would make an excellent workshop paper but its contributions are not enough for an ICLR paper. In addition, there is a lot of crucial information missing to understand and reproduce both Decoupled Scale Quantization & Efficient Selective Reconstruction.

## Decoupled Scale Quantization
According to the SMM equation in (1), the channels are independent from each other. I am not sure why then quantizing the values along the channel dimension would make any difference in the case of multiplication with $A$ & $C$. Per-state quantization of these matrices and of the states should lead to optimal quantization performance IF these matrix multiplications are in fact done in lower precision. Or the quantization scheme only for compression and matrix multiplications are done in higher precision. There are a lot of quantization details missing here that would explain the merits of the scheme.

## Efficient Selective Reconstruction
A lot of information missing here. Is this a local QAT scheme similar to OmniQuant for transformers? Are the learnt matrices just adjusted to the quantized states? Or they are also quantized using straight-through?

**Questions:**

* Many relevant references relate to LLM quantization in section 2.2, such as OPTQ, OmniQuant, OutlierSuppersion+, and etc.
* Figure 1: What exactly is learnable at block $N$ ? $\Delta$ is mentioned in the caption but is missing from the graph.
* Figure 2: is this the memory consumption during pre-fill or generation?
* Theorem 1 is very hard to follow. What does the split equation operator do? It’s never introduced. Where did the weight matrices ($W_x$ & $W_b$) come from? Please make this section more readable.
* Figure 6, the legend hides the graph
* Figure 9, the caption does not explain what the different colours are.
* It would be very useful to add a graph of the Mamba block highlighting the final precision of each tensor and the precision of each compute. Look at Figure 6  of SmoothQuant.
## Section 5.2:
  * you never introduce an equation of what block-wise reconstruction means. I understand from the references it is the $\ell_2$ loss between quantized and unquantized values but that's because I happen to know one of the cited papers well.
 * What do you learn exactly? Where is the quantization? Do you use a straight-through estimator? Do you also learn the quantization scales?

---

> ### Author Response · Authors · 2024-11-26
>
> Thank you for your careful reading of our article and providing helpful feedback. We have scrutinized the manuscript and made corresponding modification. We have expanded the introduction to include more details about SSMs and Mamba models ( e.g., the definition of $W_x, W_b$ in Theorem 1) and revised the method section to address and supplement the previously missing information (including Figure 1 and equation for finetuning).
>
> **Q1**:  **Decoupled Scale Quantization**. According to the SSM equation in (1), the channels are independent from each other. I am not sure why then quantizing the values along the channel dimension would make any difference in the case of multiplication with A & C. Per-state quantization of these matrices and of the states should lead to optimal quantization performance IF these matrix multiplications are in fact done in lower precision. Or the quantization scheme only for compression and matrix multiplications are done in higher precision. There are a lot of quantization details missing here that would explain the merits of the scheme.
>
> **R1**: During the generation stage, state caches $h_t$ are quantized at each time step to reduce memory consumption and are subsequently dequantized in the next step, which is similar to KV cache quantization. We implement a cuda kernel that load quantized $h_t{-1}$, compute SSM $h_t=A_th_{t-1}+B_tx_t$, then quantize and pack $h_t$ and store into memory.  Quantization scale **can not** be fused into $A_t$ or $C_t$ because $A_t$ is a scalar matrix  and $C_t$ only have impact on outputs $y_t$ at the current timestep.
>
> **Q2:** **Efficient Selective Reconstruction**.   Are the learnt matrices just adjusted to the quantized states? Or they are also quantized using straight-through? What do you learn exactly? Where is the quantization? Do you also learn the quantization scales?   Is this a local QAT scheme similar to OmniQuant for transformers?  You never introduce an equation of what block-wise reconstruction means.
>
> **R2:**
>
>   ESR aims to adjust parameters for the quantized states, which is a non-trivial task for SSMs. SSMs (and other linear attention mechanisms) rely on the equivalence between two computational modes: the quadratic mode for training and the recurrent mode for inference. This equivalence depends on the linear nature of the hidden-to-hidden transformations, $h_t = A_t h_{t-1} + B_t x_t$. In contrast,  RNNs, defined by $h_t = \sigma(A_t h_{t-1} + B_t x_t)$ (where $\sigma$ is a nonlinear activation function) cannot be parallelized ( known as backpropagation through time (BPTT)) . To minimize memory bandwidth utilization, we store state caches as low-bit elements and subsequently load and dequantize them before computation at the next timestep. This process introduces a new sequence transformation, defined as $h_t^q = \bar{A} Q(h_{t-1}^q) + \bar{B} x_t$, where $Q(\cdot)$ is a non-linear quantization function. It is crucial to distinguish $h_t^q = \bar{A} Q(h_{t-1}^q) + \bar{B} x_t$ from the quantized value of the original $h_t$, i.e., $Q(h_t) = Q(\bar{A} h_{t-1} + \bar{B} x_t)$. A naive approach would directly apply recurrent mode for token-by-token generation in the training. However, this method becomes computationally prohibitive for large input lengths (e.g., 2048).
>
> To address this, we propose Efficient Selectivity Reconstruction (ESR), which utilizes the parallel algorithm to compute $h_t$ across all timesteps and simulates quantization errors by **quantizing only a single step during training**. We employ straight-through estimation for the backpropagation of the quantization function $Q$.  In this process,  we freeze the linear projections corresponding to the SSM inputs $x$ and $z$, while keeping the linear projections for selective parameters $B$, $C$, and $\Delta$ learnable (account for only about 2\% of the total parameters). So far, we do not make the quantization scales learnable. Specifically, we have updated Figure 1 to highlight the learnable parameters and added a new equation to define this optimization process in Section 5.2.2
>
> **Q3:** Figure 2: is this the memory consumption during pre-fill or generation?  Figure 6, the legend hides the graph. Figure 9, the caption does not explain what the different colours are.
>
> **R3:**  Figure 2 shows the memory consumption during generation. Because of time limitation, we will modify Figure 6 and Figure 9 in the final papers.

---

> > ### Comment · Reviewer_UHAm · 2024-12-02
> >
> > Dear authors,
> >
> > Thank you for taking my questions and recommendations onboard. The revised version of the paper is much easier to read and addresses many of my initial concerns. Specifically, I now understand the importance of the _per-tensor_ quantization of the states. In addition, it is now much clearer what is learnt in the case of SGA.
> >
> > I am happy to revise my score to borderline acceptance. Whereas the quality of the paper is much improved and the investigations are well-founded, the individual contributions are not novel enough for a strong accept.

---

### Official Review · Reviewer_S8h7 · 2024-11-03

**Soundness:** 4
**Presentation:** 3
**Contribution:** 4
**Rating:** 8
**Confidence:** 5

**Summary:**

This paper explores post-training quantization (PTQ) for Mamba language models and introduces a new method called Q-Mamba. The authors identify that the bottlenecks in Mamba models arise from the large size of linear projections and state caches, which can be further optimized through model quantization. They observe that state caches contain outliers in both state and channel dimensions. Based on further analyses of this phenomenon, Q-Mamba proposes Decoupled Scale Quantization, utilizing independent scales for different dimensions. Additionally, Q-Mamba introduces a new quantization simulation approach called Efficient Selectivity Reconstruction to facilitate efficient layer-wise reconstruction using the original parallel scan algorithm. Experiments demonstrate the necessity of these methods for effective quantization of Mamba models, achieving W8A8H4 quantization for Mamba models for the first time. Experiments on memory consumption and latency are also presented.

**Strengths:**

1. The main contribution of this work is the first exploration of quantization for Mamba models. Mamba architecture has recently attracted significant attention in the research community, and this paper offers a fresh perspective on model compression. The authors identify that the bottlenecks in Mamba models stem from the large size of linear projections and state caches, highlighting the potential for further research in model compression for Mamba models.

2. Both the DSQ and ESR methods are well-motivated and specifically tailored for Mamba architectures. The DSQ method is based on the observation that states in SSMs contain outliers in both channel and state dimensions. After a theoretical analysis of this phenomenon, the paper proposes utilizing decoupled (independent) scales for these dimensions. This approach is compelling and demonstrates significant improvements compared to per-channel quantization. ESR is a novel quantization simulation method that approximates quantization noise in layer-wise reconstruction and is necessary when introducing quantization into the parallel scan algorithm. The authors also conduct empirical experiments to select fine-tuning parameters that enhance performance

3. Experiments demonstrate the effectiveness of these methods for quantization of Mamba models. Notably, this paper achieves W8A8H4 (8-bit linear projections and 4bit states) quantization for Mamba models for the first time. For example, Q-Mamba achieves a 50% reduction in memory consumption with only a 2.13% average accuracy degradation on zero-shot tasks for Mamba2-2.7B. Ablation studies also demonstrate the convenience of Q-Mamba, which requires only 128 samples and a few epochs. Experiments on memory consumption and latency are also presented.

4. This paper is clearly written, logically structured, and easy to understand.

**Weaknesses:**

1. The background section could be enhanced for clarity for readers who are not familiar with Mamba. It would be beneficial to include additional information about the differences between the Mamba-1 and Mamba-2 models.

2. Some ablation studies on hyperparameters, such as optimizer settings ( learning rate, and weight decay) should be provided to demonstrate the robustness of ESR.

3. Considering the need for fast deployment, the total time consumption for layer-wise construction should be included, particularly for the largest Mamba models.

**Questions:**

The appendix provides the results of the GPTQ[1] and SmoothQuant[2] methods on Mamba models. Could you provide more results for different quantization settings (e.g., W3A16, W4A4)? I am curious about the bit precision that can be achieved with Mamba model quantization.

[1] Xiao et al., SmoothQuant: Accurate and Efficient Post-Training Quantization for Large Language Models. ICML 2023.

[2] Frantar et al., OPTQ: Accurate Quantization for Generative Pre-trained Transformers. ICLR 2023.

---

> ### Author Response · Authors · 2024-11-26
>
> **Q1:** The background section could be enhanced for clarity for readers who are not familiar with Mamba. It would be beneficial to include additional information about the differences between the Mamba-1 and Mamba-2 models.
>
> **R1:** Thank you for your helpful feedback.  We have sigificantly modified this section in the revised paper.
>
> **Q2:** Some ablation studies on hyperparameters, such as optimizer settings ( learning rate, and weight decay) should be provided to demonstrate the robustness of ESR.
>
> **R2:** For all size models, we set the same hyperparameters. We utilize the AdamW optimizer with zero weight decay to optimize the learnable parameters in ESR. The learning rate for the learnable parameters is set to 1E-3. In Table 1, we demonstrate that different learning rates can all lead to improvements in performance.
>
> **Table 1 Impact of learning rate.**
>
> | LR             | w/O ESR | 1E-5  | 1E-4  | 5E-4  | 1E-3  | 5E-3  |
> | -------------- | ------- | ----- | ----- | ----- | ----- | ----- |
> | wiktiext (PPL) | 25.73   | 24.88 | 24.24 | 22.46 | 21.99 | 23.99 |
>
> **Q3:** Considering the need for fast deployment, the total time consumption for layer-wise construction should be included, particularly for the largest Mamba models.
>
> **R3:**  ESR only needs a single A800 80G GPU and with only 128 training samples. The training time takes about 8 hours for the Mamba2-2.7B.
>
> **Q4:** The appendix provides the results of the GPTQ[1] and SmoothQuant[2] methods on Mamba models. Could you provide more results for different quantization settings (e.g., W3A16, W4A4)? I am curious about the bit precision that can be achieved with Mamba model quantization.
>
> **R4:** Table 2 presents experimental results on various weight bit configurations for the Mamba2-370M model using the GPTQ [1] method. Compared to Transformers, Mamba models exhibit more significant performance degradation in extremly low-bit quantization. This issue may be attributed to large outliers in the inputs to the output projection layer, as shown in Figure 7.
>
> **Table 2  Experiments on various weight bits for Mamba2-370M with GPTQ[1] method.**
>
> | Mamba2-370m    | wikitext | c4      |      |
> | -------------- | -------- | ------- | ---- |
> | FP             | 14.16    | 16.95   |      |
> | w4-per-channel | 17.40    | 20.15   |      |
> | w4-g128        | 15.81    | 18.71   |      |
> | w3-per-channel | 45.11    | 46.30   |      |
> | w3-g128        | 25.19    | 27.98   |      |
> | w2-g64         | 2167.12  | 7030.40 |      |
>
> [1] Frantar et al. "OPTQ: Accurate Quantization for Generative Pre-trained Transformers", ICLR 2023

---

### Author Response · Authors · 2024-11-26

Thank you, reviewers, for your valuable feedback and constructive suggestions. Based on your comments, questions, and recommendations, we have made several revisions that significantly enhance the quality of the paper. We want to first answer the fundamental questions shared by reviewer  UHAm, tXAm, and Aixt, from the following aspects.

**(1) What we are quantizing (the hidden states) in Q-Mamba and why to quantize them.**

In this study, we emphasize that Q-Mamba focuses on the quantization of **state caches** in Mamba architectures, setting it apart from previous research on quantizing transformers and CNNs. States in Mamba are expanded to be $N$-times larger than standard activations, where $N$ represents the state dimension (128 in Mamba-2 models). Section 4 demonstrates that these state caches contribute significantly to memory consumption, particularly after weights are quantized to low-bit representations. In Section 5.1, we analyze and visualize the outlier distributions within states, revealing that this phenomenon arises from the computation of outer products between two activations, each containing outliers in distinct dimensions.   **It is important to note that the quantization of linear projections for Mamba models is not the primary contribution of this work and we present preliminary experiments on in the appendix. Additionally, Q-Mamba is orthogonal and can be combined with previous quantization methods for linear layers, such as OmniQuant, AWQ, and Mamba-PTQ.**

**(2) Why states can not be quantized directly by previous PTQ methods used in Transformers.**

**We emphasize that adjusting parameters for quantized states is a non-trivial task in block-wise reconstruction.**   SSMs (and other linear attention mechanisms) rely on the equivalence between two computational modes: the quadratic mode for training and the recurrent mode for inference. This equivalence depends on the linear nature of the hidden-to-hidden transformations, $h_t = A_t h_{t-1} + B_t x_t$. In contrast,  RNNs, defined by $h_t = \sigma(A_t h_{t-1} + B_t x_t)$ (where $\sigma$ is a nonlinear activation function) cannot be parallelized ( known as backpropagation through time (BPTT)) . To minimize memory bandwidth utilization, we store state caches as low-bit elements and subsequently load and dequantize them before computation at the next timestep. This process introduces a new sequence transformation, defined as $h_t^q = \bar{A} Q(h_{t-1}^q) + \bar{B} x_t$, where $Q(\cdot)$ is a non-linear quantization function. It is crucial to distinguish $h_t^q = \bar{A} Q(h_{t-1}^q) + \bar{B} x_t$ from the quantized value of the original $h_t$, i.e., $Q(h_t) = Q(\bar{A} h_{t-1} + \bar{B} x_t)$. A naive approach would directly apply recurrent mode for token-by-token generation in the training. However, this method becomes computationally prohibitive for large input lengths (e.g., 2048). To address this, we propose Efficient Selectivity Reconstruction (ESR), which utilizes the parallel algorithm to compute $h_t$ across all timesteps and simulates quantization errors by quantizing only a single step during training.

**The revised sections are highlighted in blue within the paper.**

- Expanded the background section to include more details definition of Mamba models and an introduction of parallel training of SSMs.
- Enhanced the explanation of ESR in Section 5.2. Speciafically we updated Figure 1 to emphasize the learnable parameters and added a new equation to define the optimization process. We have also included the pseudocode for the parallel training of quantized SSMs in the appendix to enhance understanding.
- Simplified the explanations of Theorem 1 to improve readability.

---

> ### Comment · Reviewer_UHAm · 2024-11-26
> **Revision?**
>
> As of this time, the submission does not have a revision attached. In addition, responding to our reviews on the last day of the discussion phase does not give us time to assess your responses properly.

---

> ### Author Response · Authors · 2024-11-26
>
> We sincerely appreciate your reminder and have now submitted the revised version of the paper. We understand your concern regarding the timing, and we apologize for any inconvenience this may have caused. According to the official notification, the discussion period has been extended to December 2nd (a six-day extension). Thank you again for your valuable feedback. If you have any additional questions or suggestions, please let us know

---

### Meta-Review · Area_Chair_crcZ · 2024-12-20

**Metareview:**

This paper tackles post-training quantization for Mamba models, showing notable memory savings with minimal accuracy tradeoffs. Reviewers agree that it is the first to systematically address Mamba quantization and that the techniques reduce model size and maintain reasonable performance. They appreciate the clear methodology and practical relevance. However, all note concerns about limited novelty and not enough baseline comparisons. Some also mention a need for more detailed latency analysis. Despite diffrences in final scores, the consensus is that the paper makes a meaningful engineering contribution while requiring further expansions of comparisons, tasks, and evaluations to fully demonstrate its value.

**Additional Comments On Reviewer Discussion:**

Reviewers raised concerns about: (1) lack of innovation in quantization techniques, (2) insufficient comparison with existing methods, (3) missing technical details about DSQ and ESR implementations, and (4) unclear unique challenges of quantizing SSMs vs transformers. The authors addressed these by clarifying that Q-Mamba focuses specifically on state cache quantization, explaining the technical differences between SSM and RNN training modes, and providing implementation details. One reviewer upgraded their rating from reject to borderline accept, acknowledging improved clarity while maintaining concerns about novelty. Two other reviewers maintained their original scores after the rebuttal.

---

### Decision · Program_Chairs · 2025-01-22

Reject